# How Cloud Droplet Number Concentration Impacts Liquid Water Path and Precipitation in Marine Stratocumulus Clouds—A Satellite-Based Analysis Using Explainable Machine Learning

Lukas Zipfel [1,2,*], Hendrik Andersen [1,2], Daniel Peter Grosvenor [3,4] and Jan Cermak [1,2]

1   Institute of Meteorology and Climate Research, Karlsruhe Institute of Technology (KIT), 76131 Karlsruhe, Germany; hendrik.andersen@kit.edu (H.A.); jan.cermak@kit.edu (J.C.)
2   Institute of Photogrammetry and Remote Sensing, Karlsruhe Institute of Technology (KIT), 76131 Karlsruhe, Germany
3   Met Office Hadley Centre, Exeter EX1 3PB, UK; daniel.grosvenor@metoffice.gov.uk
4   Institute for Climate and Atmospheric Science, School of Earth and Environment, University of Leeds, Leeds LS2 9JT, UK
*   Correspondence: lukas.zipfel@kit.edu

**Abstract:** Aerosol–cloud–precipitation interactions (ACI) are a known major cause of uncertainties in simulations of the future climate. An improved understanding of the in-cloud processes accompanying ACI could help in advancing their implementation in global climate models. This is especially the case for marine stratocumulus clouds, which constitute the most common cloud type globally. In this work, a dataset composed of satellite observations and reanalysis data is used in explainable machine learning models to analyze the relationship between the cloud droplet number concentration ($N_d$), cloud liquid water path (LWP), and the fraction of precipitating clouds (PF) in five distinct marine stratocumulus regions. This framework makes use of Shapley additive explanation (SHAP) values, allowing to isolate the impact of $N_d$ from other confounding factors, which proved to be very difficult in previous satellite-based studies. All regions display a decrease of PF and an increase in LWP with increasing $N_d$, despite marked inter-regional differences in the distribution of $N_d$. Polluted (high $N_d$) conditions are characterized by an increase of 12 gm$^{-2}$ in LWP and a decrease of 0.13 in PF on average when compared to pristine (low $N_d$) conditions. The negative $N_d$–PF relationship is stronger in high LWP conditions, while the positive $N_d$–LWP relationship is amplified in precipitating clouds. These findings indicate that precipitation suppression plays an important role in MSC adjusting to aerosol-driven perturbations in $N_d$.

**Keywords:** aerosol–cloud–precipitation interactions; cloud droplet number concentration; machine learning; marine stratocumulus; remote sensing; satellite observations





## 1. Introduction

Marine stratocumulus clouds (MSC) are known as the most common cloud type, covering more than 20% of the global oceans in the annual mean with local coverage over subtropical and midlatitude oceans sometimes above 50% [1]. MSC have a strong impact on the Earth's radiative budget by increasing the planetary albedo, as they are brighter than the ocean surface below. Comparatively small changes in the radiative properties of such marine boundary layer clouds can, therefore, lead to large changes in global shortwave reflectivity [2,3]. Changes in cloud properties in response to aerosol emissions and climate change can, thus, lead to large uncertainties in predictions of future climate [4–6].

The cloud shortwave radiative effect (CRE$_s$) of MSC is dependent on the clouds' properties, such as effective droplet radius ($r_e$), cloud droplet number concentration ($N_d$),

liquid water path (LWP), and cloud fraction (CLF), which in turn depend on environmental conditions. $N_d$ is related to the availability of aerosols, which function as cloud condensation nuclei (CCN) during cloud formation. Increases in $N_d$ with constant liquid water will lead to a lower $r_e$, resulting in an increase in the $CRE_s$ known as the Twomey effect [7]. In addition to this immediate effect on the $CRE_s$, subsequent effects from aerosol–cloud–precipitation interactions (ACI) through resulting cloud adjustments further alter the radiative forcing.

For polluted clouds (characterized by higher CCN), the more numerous and smaller droplets lead to subsequent LWP adjustments within the cloud through two counteracting pathways: precipitation suppression and entrainment feedback [8,9]. The initial droplet formation process in MSC through condensation of liquid water on CCN can produce droplets up to a size of ∼20 μm [1]. Further growth to droplets large enough to precipitate is realized through coalescence (the merging of cloud droplets), either by collision of multiple small droplets (termed "autoconversion" [10]) or by a larger droplet that collects smaller droplets ("accretion"). Under conditions with elevated $N_d$ and a resulting decrease in $r_e$ the process of collision–coalescence is inhibited due to its strong dependence on the droplet size ([1]). The subsequent decrease in precipitation is expected to lead to an increase in LWP and cloud lifetime [11] due to a less stable boundary layer and the weakened cloud liquid water sink [12]. However, the extent to which the process of precipitation suppression impacts clouds is unclear. Previous studies that use observations of natural and anthropogenic aerosol emission events to estimate the impact of indirect ACI on clouds suggest there is no or only a negligible LWP adjustment in clouds as a response to $N_d$ perturbations [9,13], calling into question the impact of precipitation suppression on a global scale. In contrast, in a recent study, Gupta et al. [14] showed the importance of precipitation suppression in environments with elevated levels of aerosol for the Southeast Atlantic Ocean (SEA) with observation data acquired in the NASA ObseRvations of Aerosols above CLouds and their intEractionS (ORACLES) campaign [15,16]. In addition to the suppression of precipitation, the reduced droplet radius is also hypothesized to lead to a decrease in droplet sedimentation speed [17,18] and an increased evaporation–entrainment rate at the cloud top acting as a LWP sink [19–21], which is further enhanced in less stable situations with a lower humidity above the cloud top.

Accordingly, by way of the two opposing processes described above, the cloud LWP adjustment in response to $N_d$ perturbations could potentially be positive, negative, or even absent [8]. Previous studies have shown that the LWP response to $N_d$ perturbations is dependent on the cloud state [3,22] with findings of a stronger increase in LWP with $N_d$ in precipitating clouds likely attributed to precipitation suppression [23]. If precipitation suppression is the dominating process through which $N_d$ controls the LWP sinks, it is to be expected that there is a decrease in precipitation with increasing $N_d$. On the other hand, a decrease in LWP in non-precipitating clouds with increasing $N_d$ may be indicative of enhanced entrainment feedback. The objective of this study is to analyze the impact of changes in $N_d$ on precipitation and liquid water path in MSC by utilizing statistical machine learning models and data from satellite observation. While previous studies (e.g., [8,24,25]) have presented the $N_d$–LWP relationship using satellite data, it has proved very difficult or impossible to disentangle the impact of changes in $N_d$ alone as a causative factor due to various other confounding factors. In this work, the use of machine learning models allows these confounding factors to be controlled for, enabling the analysis of the impact of $N_d$ in isolation. This represents a step-change in the ability to use observations to quantify ACI effects and provides a robust metric for testing and improving climate models.

## 2. Materials and Methods

This study is conducted for five marine regions that each cover 10° by 10° and are characterized by a high annual occurrence of stratocumulus clouds as defined by Klein and Hartmann [26]. The regions are named according to the closest eastward coastline as displayed in Table 1. A machine learning model is fitted in each region based on a

combination of satellite observations and reanalysis model data in order to asses the impact of changes in cloud droplet number concentration ($N_d$) on precipitation and LWP, and factors controlling these relationships.

**Table 1.** Locations of the study domains with the total number of observations (N) and the explained variability for the PF ($R^2_{pf}$) and LWP ($R^2_{lwp}$) models.

| Region Name | Latitude | Longitude | N | $R^2_{pf}$ | $R^2_{lwp}$ |
|---|---|---|---|---|---|
| Australia | 25° S–35° S | 95° E–105° E | 16,504 | 0.63 | 0.61 |
| California | 20° N–30° N | 120° W–130° W | 18,919 | 0.71 | 0.65 |
| Canaries | 15° N–25° N | 25° W–35° W | 8431 | 0.65 | 0.67 |
| Namibia | 10° S–20° S | 0°–10° E | 20,337 | 0.68 | 0.66 |
| Peru | 10° S–20° S | 80° W–90° W | 23,512 | 0.63 | 0.66 |

*2.1. Data*

The dataset used here is a refined version of the data used in Zipfel et al. [23]. Satellite observations of various cloud properties are taken from the CALIPSO-CloudSat-CERES-MODIS Merged Release B1 (C3M) product, which is available for July 2006 to April 2011. The C3M dataset is based on the Clouds and the Earth's Radiant Energy System (CERES) with a resolution of ∼20 km [27]. Observations from multiple additional sensors (Cloud-Aerosol Lidar and Infrared Pathfinder Satellite Observation (CALIPSO), Cloudsat, and Moderate Resolution Imaging Spectroradiometer (MODIS)) are collocated for each of these CERES "footprints". A footprint, therefore, represents an individual observation with data from all four instruments combined. Each CERES footprint is further segmented into a maximum of 16 cloud groups (distinguished as unique entities as seen from above) and up to 6 cloud layers (distinguished vertically) by utilizing the high resolution and vertical profiles provided by CALIPSO and CloudSat.

Only CERES footprints with a single cloud layer are used in this study to exclude the influence of superimposing clouds. Additionally only observations of low-level clouds are selected by limiting the cloud top height (CTH) to 3 km [28]. CTH is defined as the median of all cloud groups in a CERES footprint.

To analyze the impact of the cloud droplet number concentration on precipitation formation, the CloudSat precipitation flag is used to calculate the precipitation fraction (PF). The precipitation flag consists of four possible classes (no precipitation, liquid, solid, or drizzle) and provides this classification for each cloud group. The PF is calculated for each CERES footprint and is defined as the number of cloud groups where any form of precipitation is detected by CloudSat, divided by the total number of cloud groups [23]. Drizzle is the prevalent class of precipitation detected over all regions (>88% of precipitating cloud groups). Only a total of two observations of solid precipitation are found in the Peru region.

Using MODIS retrievals of the effective cloud-droplet radius ($r_e$), the cloud optical depth ($\tau_c$), the cloud-top temperature, and the cloud-top pressure, the cloud-droplet number concentration ($N_d$) is calculated according to Grosvenor et al. [29]:

$$N_d = \frac{\sqrt{5}}{2\pi k} \sqrt{\frac{f_{ad}\, c_w\, \tau_c}{Q_{ext}\, \rho_w\, r_e^5}} \tag{1}$$

where $k = 0.8$, $f_{ad} = 0.66$, and $Q_{ext} = 2$. While the overall uncertainty in the calculated $N_d$ is estimated to amount to around 78% at the native MODIS resolution of 1 km$^2$ [29], this is likely reduced for the footprint averages utilized here. The accuracy of the $N_d$ calculation was further improved by utilizing the $r_e$ and the $\tau_c$ provided by the CERES enhanced cloud algorithm. The enhanced cloud algorithm makes use of the CALIPSO and CloudSat cloud height and cloud mask retrievals in addition to the MODIS reflectances to obtain $r_e$ and $\tau_c$ [27], reducing the number of cases where default values for observations outside the lookup table are used in the standard CERES algorithm. Furthermore, any instances where

$r_e$ or $\tau_c$ are below 4 are removed from the dataset due to the associated uncertainties for retrievals in this range [30]. Finally, only the 1st–99th percentiles for $N_d$ are used in order to remove any remaining outliers possibly introduced by measurement errors.

Information on the background environment is provided to the model through the meteorological reanalysis data taken from the ERA5 dataset available through the European Centre for Medium-Range Weather Forecasts (ECMWF) at a 0.25° × 0.25° resolution on an hourly basis [31,32]. Data for mean sea-level pressure (MSL), sea-surface temperature (SST), air temperature, relative humidity (RH), u, and vs. wind components and vertical velocity are collocated with the C3M data. The pressure levels in the ERA5 data that are closest to the lower and upper cloud boundaries are then chosen based on the cloud-base height (CBH) and CTH from CALIPSO. For each CERES footprint, these pressure levels are then used to select the temperature, RH and wind components below the cloud, and the RH and wind components above the cloud (Table 2). Additionally, the 2 m air temperature and the temperature at 700 hPa from ERA5 are used to calculate the estimated inversion strenght (EIS) according to Wood and Bretherton [33], assuming a surface pressure of 1010 hPa.

**Table 2.** Overview of the variables used in the machine learning models.

| Variable Name | Abbreviation | Origin |
|---|:---:|:---:|
| Temperature below cloud | $T_{bc}$ | ERA5 |
| Vertical velocity below cloud | $w_{bc}$ | ERA5 |
| Winds below cloud | $u_{bc}/v_{bc}$ | ERA5 |
| Winds above cloud | $u_{ac}/v_{ac}$ | ERA5 |
| Relative humidity below cloud | $RH_{bc}$ | ERA5 |
| Relative humidity above cloud | $RH_{ac}$ | ERA5 |
| Mean sea level pressure | MSL | ERA5 |
| Sea surface temperature | SST | ERA5 |
| Estimated inversion strength | EIS | ERA5 |
| Cloud top height | CTH | CALIPSO |
| Precipitation fraction | PF | CloudSat |
| Cloud droplet number concentration | $N_d$ | MODIS |
| Liquid water path | LWP | AMSR-E |
| Rain water content [1] | RWC | AMSR-E |

[1] RWC not used to predict LWP.

The liquid water path (LWP) and the rain water content (RWC) are obtained from the Level-2B precipitation product Version 3 of the Advanced Microwave Scanning Radiometer-Earth Observing System (AMSR-E) sensor aboard the Aqua satellite. To avoid the risk of an introduction of a pseudo-relationship in the $N_d$ and LWP retrievals due to correlated errors when both are based on the MODIS sensor, the AMSR-E product is utilized as an independent source of LWP measurement. The data are provided with a resolution of 5 km across track and 10 km along track [34]. To maintain a similar spatial scale for all satellite observations, the mean LWP and the mean RWC is calculated for each CERES footprint based on the five AMSR-E pixels closest to the center of that footprint as in [23].

### 2.2. Models

Gradient Boosting Regression Tree (GBRT) models are trained to predict PF and LWP in all regions. GBRTs are able to capture non-linear relationships while considering the interaction effects between the predictor parameters, and do not depend on the input data having a specific distribution [35]. GBRT and other tree based machine learning models have been utilized to analyse low-level clouds and ACI in the past [23,36–39]. The input variables for the machine learning models are displayed in Table 2. The PF model utilizes all variables as input features, while the LWP model does not use the RWC. This was done to reduce the impact of confounding errors in those models, as both LWP and RWC are retrieved from the AMSR-E sensor, which is known to have problems separating LWP and RWC especially in situations with low amounts of liquid water [40]. Shapley additive

explanation (SHAP) values are calculated for each individual model and used to analyze the effect of changes in $N_d$ on the predicted cloud parameters (PF and LWP).

SHAP values quantify the contribution of each predictor variable (model feature) to the model prediction [41,42]. The sign of a feature's SHAP value corresponds to a decrease (negative SHAP value) or increase (positive SHAP value) the model prediction (relative to the mean prediction value) in response to a feature's observed value. Therefore, each model prediction for an individual observation can be described as the sum of the mean model prediction (defined as the mean of the predictand variable provided to the model for training) and the SHAP values for all model features related to that observation. The SHAP values themselves are the sum of the main effects, which are attributed to a single feature (the current main feature) and the interaction effects which account for the impact of the corresponding secondary features. Interaction effects can be analyzed to explore the relationship between multiple model features and their impact on the model prediction. They are defined as the change in the SHAP value of the main feature that occurs when the secondary feature is removed.

Prior to the analysis of the GBRT models' SHAP values, hyperparameter tuning was performed to optimize the explained variability ($R^2$, Table 1) and minimize the Root Mean Squared Error (RMSE) by utilizing a training and a test dataset, which, respectively, had 70% and 30% of the data assigned randomly. The hyperparameters for the models have been adjusted following a grid search of three parameters that have the strongest impact on model performance (learning rate = 0.05, number of estimators = 1000, maximum tree depth = 3). Additionally, the minimum number of samples per leaf was set to 50 to help prevent the models from overfitting.

## 3. Results

An overview of the five study regions and the spatial distribution of the grid cell averages for $N_d$, PF, and LWP is shown in Figure 1. All regions display a gradient from the east (closer to the coastline) towards the open ocean in the west with decreasing $N_d$ and increasing PF and LWP, indicating a negative relationship of $N_d$ with PF and LWP. This gradient is less pronounced for the Canaries and the Australian region (Figure 1b,c), likely due to the larger distance to the coast and the prevailing wind direction resulting in a reduced influence of continental aerosol emissions. The continental influence is also evident from the maxima in observed $N_d$, which are highest in the Californian and Namibian region (indicated by the dark brown shading, Figure 1a,c), while the Canaries and the Australian region show markedly lower $N_d$ maxima (light brown shading, Figure 1b,c). In the following Sections 3.1 and 3.2, the $N_d$–PF ($N_d$–LWP) relationship is analyzed using machine learning, specifically exploring the influence of LWP (PF).

### 3.1. Precipitation Fraction

The regional machine learning models are able to explain 63–71% of the observed variability in PF ($R^2$ shown in Table 1) in the independent test data, displaying a similar performance across the different regions. Figure 2 shows the SHAP values for $N_d$ ($N_d$–SHAP) for the models predicting PF across the five regions. Each dot represents a single observation and is colored by LWP. All five regions are characterized by a decrease in predicted PF with increasing $N_d$ (Figure 2a–e). This negative relationship is characterized by a steep decrease in $N_d$–SHAP for the lower spectrum of the $N_d$ distribution (approximately 0–200 droplets cm$^{-3}$), which then stagnates at higher $N_d$. Therefore, increases in $N_d$ above $\sim$200 cm$^{-3}$ do not lead to a further substantial decrease in PF.

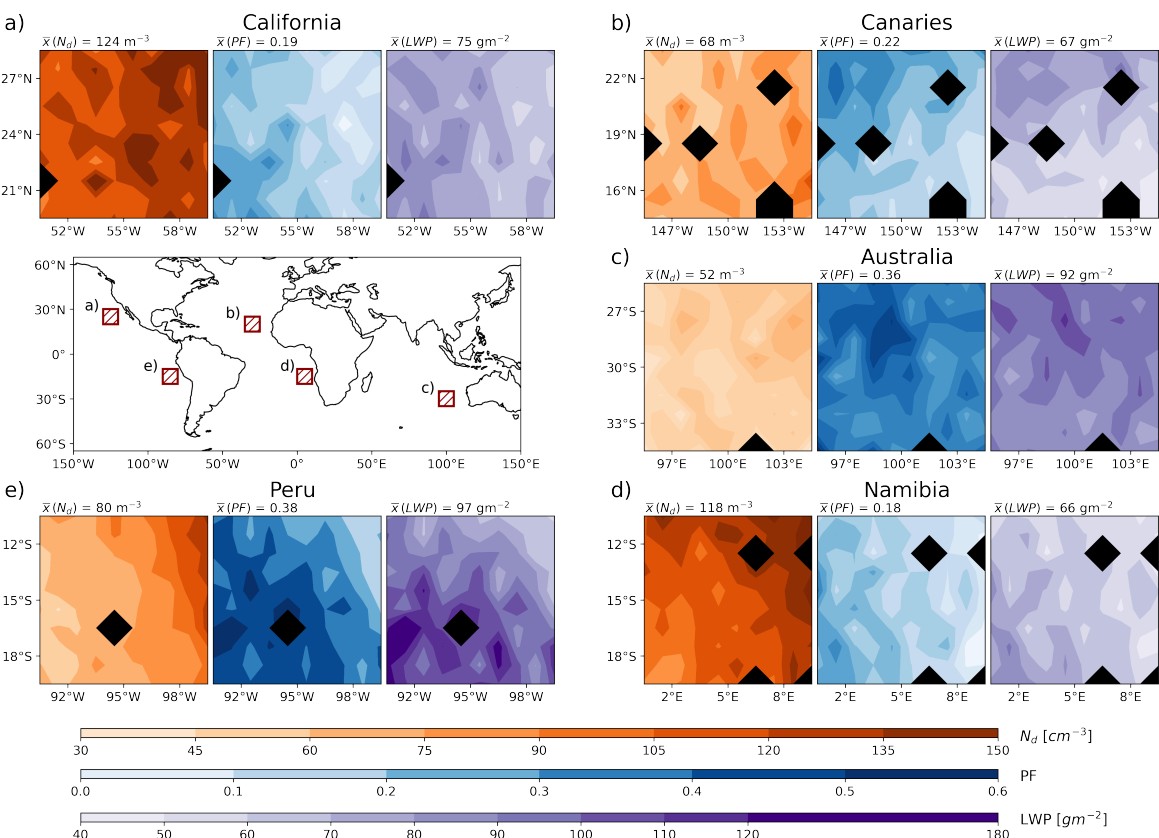

**Figure 1.** Overview of the five regions analyzed in this study. Contour plots (**a**–**e**) show the grid cell mean $N_d$ (brown shading), PF (blue shading), and LWP (purple shading) for each region based on data aggregated on a 1° by 1° grid for the years of 2006–2011. Grid cells with less than five observations are shown in black. Regional averages are displayed above the contour plots. For reference, each region (**a**–**e**) is additionally displayed as a red rectangle in the global map.

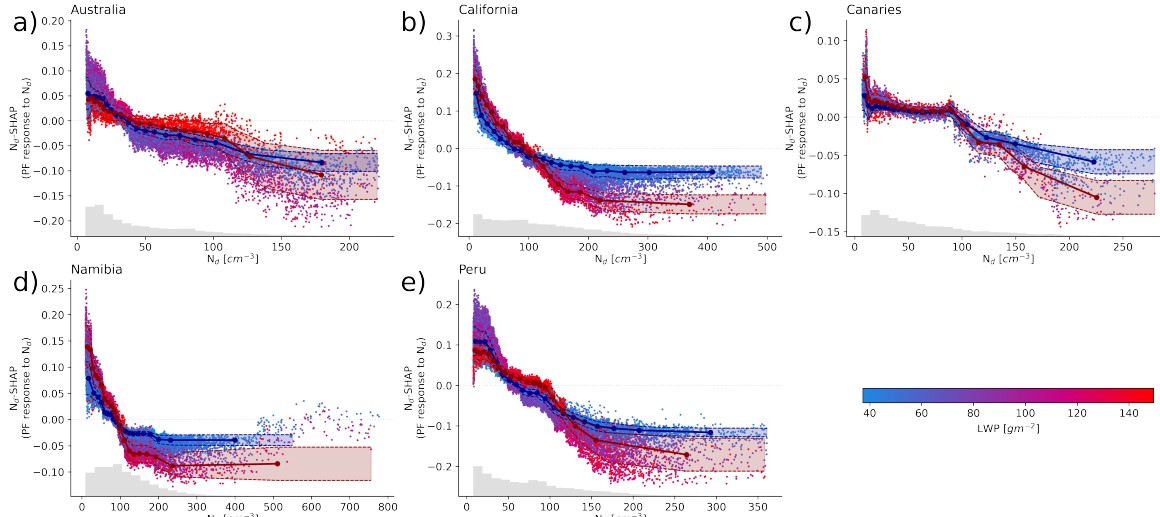

**Figure 2.** Scatter plots for the five stratocumulus regions (**a**–**e**) showing the effect of changes in $N_d$ on PF using SHAP values. Each dot represents a single observation. The LWP for each observation is indicated by color increasing from blue to red. The interconnected dots show the median $N_d$–SHAP value for 20 equal sample bins across $N_d$ for the 1st (blue) and 4th (red) LWP quartiles with one standard deviation indicated by the shaded areas. Shown in grey at the bottom of each panel is a histogram of the distribution of observed $N_d$.

To analyze the impact of LWP as a secondary model feature on the $N_d$–SHAP, we compare the observations based on the 1st and 4th quartile of LWP. The average LWP for the lower quartile is close to 40 gm$^{-2}$ for all regions, while the average LWP for the upper quartile is ∼160 gm$^{-2}$ for the Australian and Peruvian region and in the range of 120–130 gm$^{-2}$ for the Canaries as well as the Californian and Namibian region. In order to better display the predicted PF response to possible interactions with LWP, the lower and upper LWP quartiles are split into 20 equal $N_d$ sample bins (ESB). The interconnected dots in Figure 2 show the median $N_d$–SHAP value of the ESB, with shading indicating one standard deviation. The ESB lines are colored according to their respective LWP quartile (lower quartile—blue, upper quartile—red). In all regions, with the exception of Australia (Figure 2b–e), there is a discernible difference in the predicted PF response to $N_d$ between the low (blue) and high (red) LWP cases, suggesting an amplified sensitivity of PF to $N_d$ in high-LWP environments and a dampened $N_d$–PF sensitivity for below-average LWP.

To test whether these observed differences in $N_d$–PF sensitivity are actually attributed to LWP in the machine learning model, interaction effects are analyzed. The interaction effects for $N_d$ and LWP are depicted in Figure 3 and colored by LWP. Here, the interaction effects are defined as the change in the PF response to an $N_d$ value when LWP is present as a model predictor versus a model without LWP. Therefore, it quantifies how sensitive the predicted PF response to $N_d$ is to the secondary feature LWP. The interaction effects clearly show a separation of the PF response to $N_d$ by LWP. In low LWP conditions (blue dots), interaction effects tend to increase with $N_d$, whereas the opposite is the case for high LWP conditions (red dots), which becomes especially apparent in the diverging pattern for $N_d > 100$. This shows that in the machine learning model, the predicted PF response to $N_d$ is modulated by LWP where the negative $N_d$–PF sensitivity is amplified in high LWP conditions (adding the negative $N_d$–LWP interaction effect relationship to the negative $N_d$–PF relationship), but dampened in low LWP conditions (adding the positive $N_d$–LWP interaction effect relationship to the negative $N_d$–PF relationship). In the Californian, Canarian, Namibian, and Peruvian region (Figure 3b–e), observations characterized by a high LWP show a negative impact on the predicted PF as opposed to observations with a low LWP. This suggests that in the machine learning model the differences in $N_d$–PF sensitivity for high and low LWP situations are in fact caused by LWP. The distinction between high and low LWP cases is not as clear for the Australian region, however (Figure 2a). Similarly, in Figure 3a, the LWP interaction effects on $N_d$–SHAP are not as clearly separable for low and high LWP cases as for the other regions. This may be attributed to the lack of high $N_d$ observations in the region.

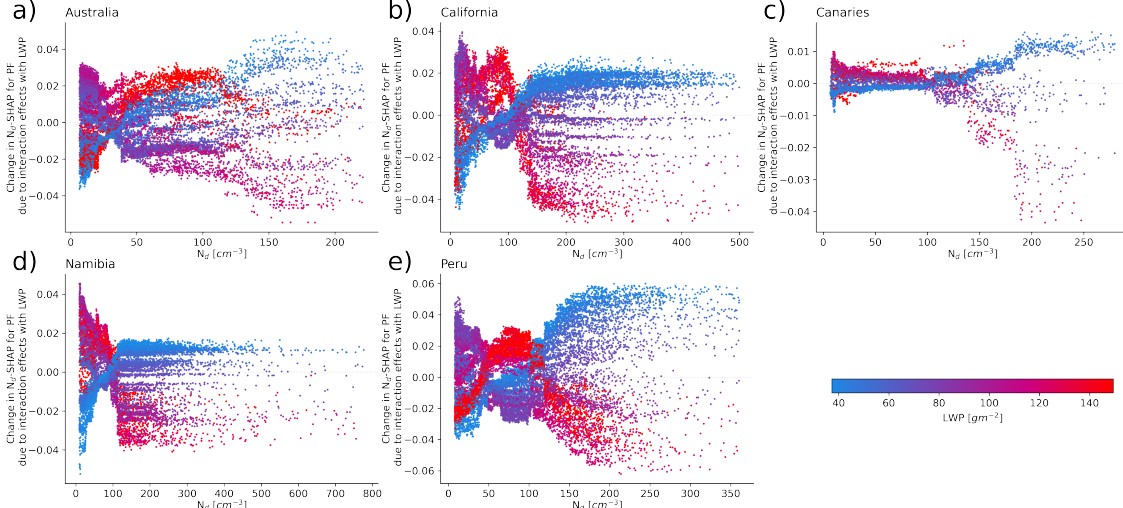

**Figure 3.** Scatter plots for the five stratocumulus regions (**a–e**) showing the interaction effects for $N_d$ and LWP when predicting PF. Each dot represents a single observation. The LWP is indicated by color increasing from blue to red.

To quantify the $N_d$–PF relationship and enable an inter-regional comparison despite marked differences in the $N_d$ distribution, two distinct $N_d$ bins are defined: (i) pristine conditions where $20 < N_d < 50$ and (ii) polluted conditions where $150 < N_d < 250$. The sensitivity of PF ($S_{PF}$) to changes in $N_d$ is then defined as the difference in the median SHAP value of the polluted bin minus the pristine bin. To assess the impact of the LWP control on the $N_d$–PF relationship suggested above (Figures 2 and 3), the 1st and 4th quartile of LWP are used to calculate $S_{PF}$ for low ($S^l_{PF}$) and high ($S^h_{PF}$) LWP conditions, respectively. $S_{PF}$ values for all regions are displayed in Figure 4 with color indicating the LWP conditions. All five regions are characterized by a negative $S_{PF}$ with an inter-regional average of $-0.13$. A negative $S_{PF}$ indicates a decrease of the predicted PF with increasing $N_d$ that is attributed solely to $N_d$. As SHAP values are always given in units of the model's predictions, they have the advantage of simplifying the interpretation of the physical impact of $S_{PF}$. Accordingly, a $S_{PF}$ of $-0.13$ refers to an $N_d$-driven median reduction of PF by 13 percentage points from pristine to polluted conditions (e.g., a median PF of 0.40 in pristine and 0.27 in polluted conditions). Throughout all five regions, high LWP conditions display a stronger negative $N_d$–PF sensitivity ($S^h_{PF} < S^l_{PF}$), with an inter-regional average $S^h_{PF}$ of $-0.16$, compared to an average $S^l_{PF}$ of $-0.10$. The Californian ($S^h_{PF} = -0.22$, $S^l_{PF} = -0.11$) and Namibian ($S_h = -0.16$, $S_l = -0.08$) regions show the largest differences in sensitivities, with $S^h_{PF}$ values twice as large as $S^l_{PF}$.

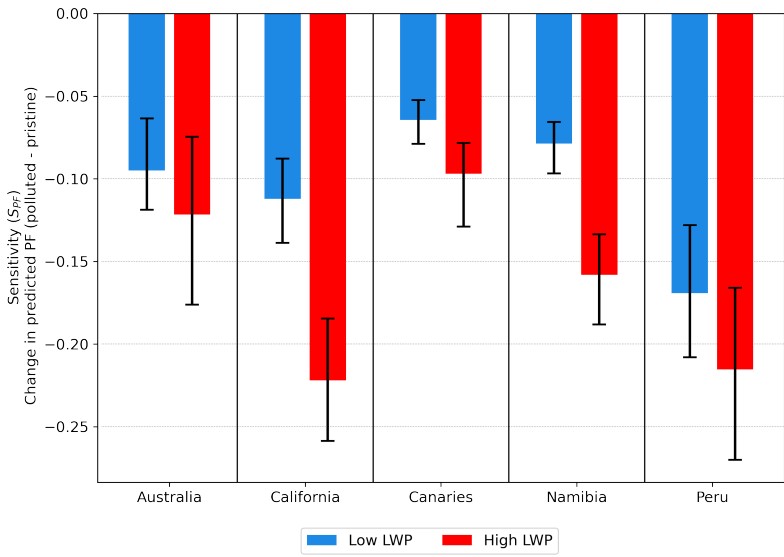

**Figure 4.** Sensitivity of PF ($S_{PF}$) to changes in $N_d$ defined as the difference in the bin median Nd-SHAP values between the $20 < N_d < 50$ and $150 < N_d < 250$ bins. Error bars show the range of the differences between the 25th and the 75th percentile of both $N_d$ bins. Values are shown for both low LWP ($S^l_{PF}$, blue) and high LWP ($S^h_{PF}$, red) conditions.

## 3.2. Liquid Water Path

Similar to the previous section, the machine learning models predict LWP equally well across the five regions with an explained variability in the range of 61–67% (Table 1) in the independent test data. Figure 5 shows the SHAP values for $N_d$ for the models predicting LWP akin to Figure 2, but with each dot colored by PF. The models for all regions show a positive $N_d$–LWP relationship (Figure 5a–e) as the SHAP values increase with higher $N_d$. The increase in predicted LWP is larger at low $N_d$ values, whereas at higher $N_d$ (i.e., $N_d > 100$), LWP is less sensitive to further increases in droplet number, which has been shown for the Namibian stratocumulus region in previous work [23]. This saturation effect is consistent across all regions except the Canaries, where the predicted LWP is shown to increase with $N_d$ even for $N_d > 100$. Aside from the Canaries, the saturation effect seems to be independent of the large inter-regional differences observed in the distribution of $N_d$.

It is worth noting that the Californian and Peruvian regions (Figure 5b,e) display a slight decrease in LWP in high $N_d$ conditions for non-precipitating clouds (blue dots), which may be indicative of entrainment feedback.

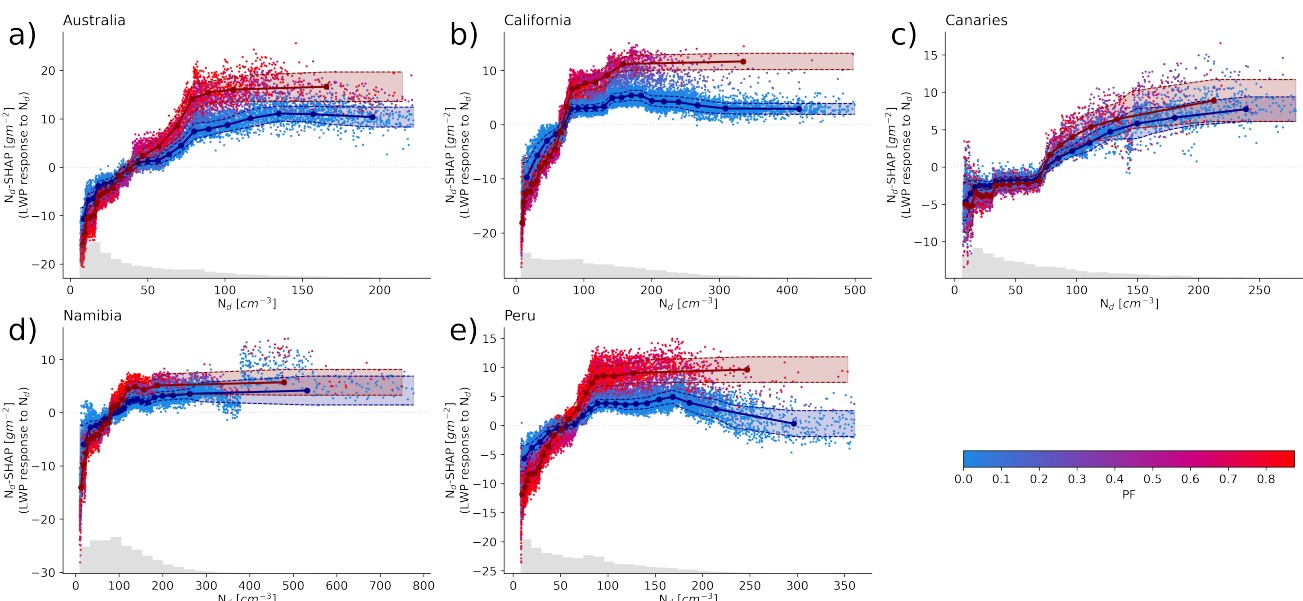

**Figure 5.** Same as Figure 2 but for models predicting LWP. Scatter plots for the five stratocumulus regions (**a**–**e**) showing the effect of changes in $N_d$ on LWP using SHAP values. Each dot represents a single observation. The PF for each observation is indicated by color increasing from blue to red. The interconnected dots show the median $N_d$–SHAP value for 20 equal sample bins across $N_d$ for the 1st (blue) and 4th (red) PF quartiles with one standard deviation indicated by the shaded areas. Shown in grey at the bottom of each panel is a histogram of the distribution of observed $N_d$.

All regions show a higher sensitivity of LWP to changes in $N_d$ for precipitating clouds (upper quartile of PF, red dots) when compared to non-precipitating clouds (lower quartile of PF, blue dots). This difference is most pronounced for the Californian and Peruvian regions (Figure 5b,e) and is driven by the opposing interaction effects of PF in precipitating and non-precipitating clouds (Figure 6). In all five regions these interaction effects lead to an increase in the predicted LWP with increasing $N_d$ in precipitating clouds (red dots), while non-precipitating clouds (blue dots) are characterized by a decrease in the predicted LWP. Using the same definition of pristine and polluted conditions as in Section 3.1, the sensitivity of LWP ($S_{LWP}$) to changes in $N_d$ is calculated as the difference in the median SHAP value for the polluted bin minus the pristine bin. Similar to Section 3.1, the 1st and 4th quartile of PF are used to asses the impact of PF on the $N_d$–LWP relationship by calculating the LWP sensitivity for non-precipitating clouds ($S_{LWP}^l$) and precipitating clouds ($S_{LWP}^h$). Figure 7 shows $S_{LWP}$ for all regions with color indicating the cloud state (non-precipitating—blue, precipitating—red). All five regions display a positive $S_{LWP}$ with an inter-regional average of 12 gm$^{-2}$ more liquid water in polluted clouds when compared to pristine conditions. This relationship is further enhanced in precipitating clouds with an average $S_{LWP}^h$ of 15 gm$^{-2}$ and weakened under non-precipitating conditions ($S_{LWP}^l = 8$ gm$^{-2}$). While the impact of PF on the $N_d$–LWP relationship can be observed in all regions, it is especially pronounced in the Californian ($S_{LWP}^h = 21$ gm$^{-2}$, $S_{LWP}^l = 9$ gm$^{-2}$) and Peruvian ($S_{LWP}^h = 15$ gm$^{-2}$, $S_{LWP}^l = 7$ gm$^{-2}$) regions.

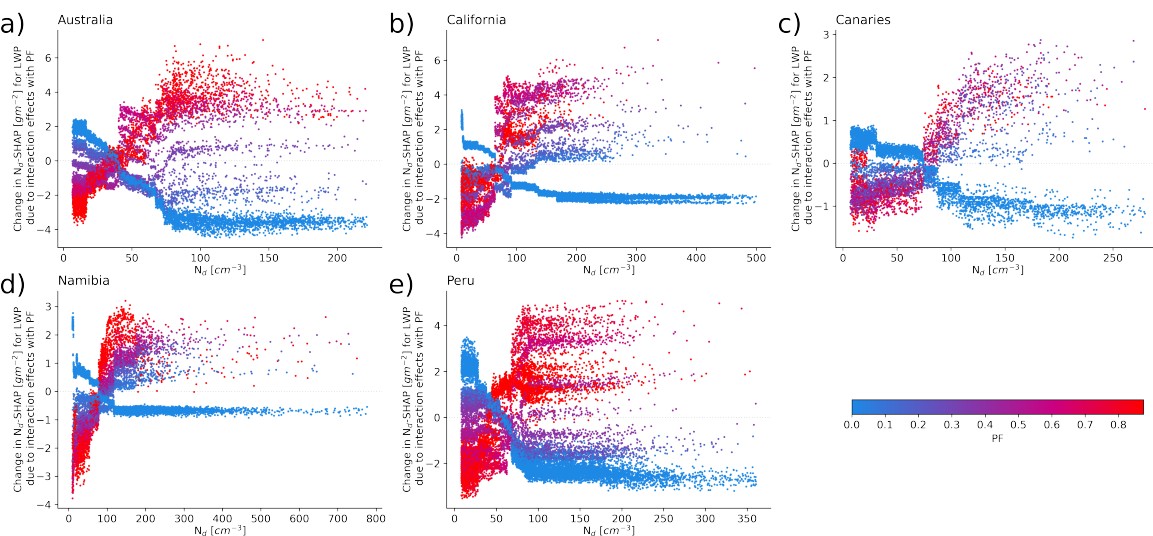

**Figure 6.** Same as Figure 3 but for models predicting LWP. Scatter plots for the five stratocumulus regions (**a–e**) showing the interaction effects for $N_d$ and PF when predicting LWP. Each dot represents a single observation. The PF is indicated by color increasing from blue to red.

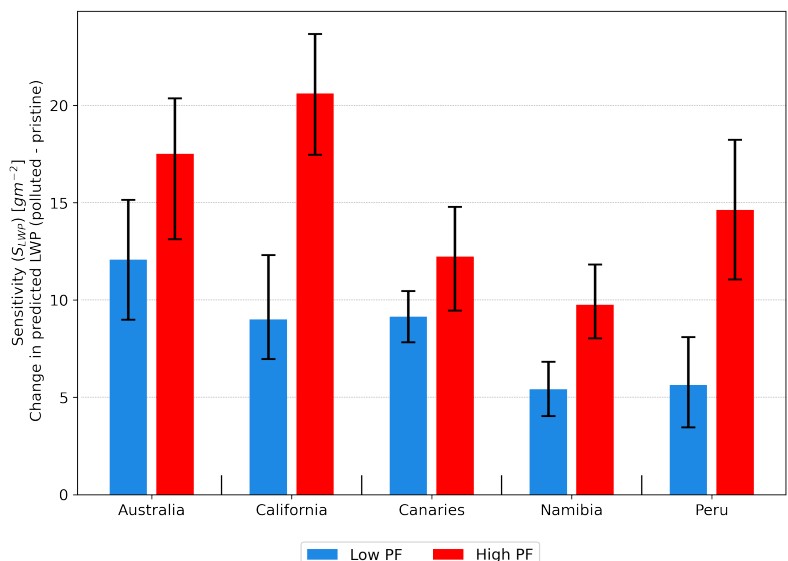

**Figure 7.** Same as Figure 4 but for models predicting LWP. Sensitivity of LWP ($S_{LWP}$) to changes in $N_d$ defined as the difference in the bin median Nd-SHAP values between the $20 < N_d < 50$ and $150 < N_d < 250$ bins. Error bars show the range of the differences between the 25th and the 75th percentile of both $N_d$ bins. Values are shown for both low PF ($S_{LWP}^l$, blue) and high PF ($S_{LWP}^h$, red) conditions.

## 4. Discussion

Previous studies examining the relationship between cloud $N_d$ and precipitation often utilize a measure of precipitation susceptibility defined as the negative fractional change in precipitation with a fractional increase in $N_d$ first introduced by Feingold and Siebert [43]. Precipitation susceptibility may be based on different proxies for precipitation, such as the rain rate, the probability of rain, and the rain intensity [44]. Here, we compare the findings of the precipitation susceptibility based on the probability of rain ($R_0$) to $S_{PF}$ based on PF. The probability of rain as defined in Wang et al. [45] lends itself to comparison with PF as both measures are calculated by relating the number of precipitating clouds to the total number of clouds observed. $R_0$ is found to be variable in strength depending on the cloud thickness ($H$) [46,47], but is reported to be strictly positive for MSC in previous

works [44,46–48] when considering the LWP range covered by the dataset analyzed here. A positive $R_0$ indicates a decrease in the probability of rain with increasing $N_d$ [43], which is in agreement with the findings for $S_{PF}$ presented here. The influence of LWP on $R_0$ is not described as unambiguously in the literature. While Terai et al. [44,46] suggested a decreasing $R_0$ in MSC with increasing $H$, Jung et al. [47] showed that there is an increase in $R_0$ with increasing $H$ in MSC up to intermediately thick clouds and Gupta et al. [14] reported an overall increase in $R_0$ with increasing $H$ especially in high $N_d$ environments. An increase in $R_0$ with $H$ agrees well with the finding of a more negative $S_{PF}$ in high LWP MSC, as both indicate a stronger decrease in the occurrence frequency of precipitation with increasing $N_d$ in clouds with more liquid water available.

Recent studies looking to analyze the $N_d$–LWP relationship globally based on MODIS satellite obseravtions find a positive sensitivity at low $N_d$ and a negative sensitivity at higher $N_d$ ($N_d > 30$), the latter of which dominates the overall correlation, leading to a decrease of LWP with increasing $N_d$ [8,25]. In contrast to this, a general pattern of LWP increasing with $N_d$ has been reported in previous studies based on models [22,49–51] as well as observation data [23,52,53], which is consistent with the findings of a positive but non-linear $N_d$–LWP relationship here. Only the Californian and Peruvian stratocumulus regions show a slight decrease in LWP under high $N_d$ conditions in non-precipitating clouds, which may be attributed to the influence of evaporation–entrainment feedback, especially since the two regions show a low average for the relative humidity above cloud (California avg. $RH_{ac} = 29\%$, Peru avg. $RH_{ac} = 20\%$). However, as opposed to the findings of Gryspeerdt et al. [8], this decrease only occurs above a higher $N_d$ threshold ($N_d > 150$), is much smaller in magnitude, and is unable to reverse the overall positive $N_d$–LWP relationship. The findings here are closer to those displayed in Ackerman et al. [17], who used large eddy simulations that indicate an LWP increase with increasing $N_d$ up to $\approx$60–100 cm$^{-3}$, depending on the relative humidity above cloud. At this point precipitation may be mostly suppressed, which could then lead to the negative impact of evaporation entrainment on LWP becoming dominant, especially under conditions with drier air above the clouds.

The observation of a decreasing PF (especially for high LWP clouds) with increasing $N_d$ along with an increasing LWP (especially in drizzling clouds) shown here can be explained by the process of precipitation suppression [11]. Clouds characterized by higher LWP that are precipitating or have the potential to develop precipitation are more susceptible to precipitation suppression. However, low LWP clouds are unable to form precipitation due to LWP acting as the limiting factor and hence there can be no precipitation suppression. This tendency of LWP enabling precipitation and $N_d$ regulating precipitation has also been described by Lu et al. [54] albeit in a more general manner for a larger range of LWP than is observed in MSC.

## 5. Conclusions

In this study, explainable machine learning models (in the form of GBRTs) were trained to predict PF and LWP over five marine areas characterized by a high occurrence of MSC. SHAP values were used to analyze the statistical relationship between $N_d$, PF, and LWP in order to provide a better understanding of cloud adjustments to aerosol perturbations. The main findings are as follows:

- The GBRT models are able to explain ∼60–70% of the variability (Table 1) in LWP and PF in the five regions considered here.
- With increasing $N_d$, an overall decrease in PF and increase in LWP is found in all five regions. The decrease in PF is amplified in high LWP clouds and the increase in LWP is stronger for precipitating clouds.
- The process of precipitation suppression is likely responsible for the observed sensitivity of PF and LWP to changes in $N_d$.

- Evaporation–entrainment feedback may be responsible for a decrease in LWP in non-precipitating clouds under high $N_d$ conditions in the Californian and Peruvian region.

The machine learning framework applied in this work explicitly considers the entire $N_d$–PF-LWP system and related environmental parameters on a fine spatial scale, thereby yielding more dependable results compared to approaches considering only the binary relationship between cloud parameters. However, even the comparatively high spatial resolution of the dataset used here is still too coarse to resolve processes on the cloud formation scale. In order to further improve the models' performance, an increased spatial resolution for the input data—especially the thermo-dynamic parameters derived from ERA5—would be desirable. Nevertheless, it was possible to demonstrate that the displayed approach is capable of isolating individual processes and is able to account for non-linearity, while also incorporating interactive effects between multiple parameters. Accordingly, the findings discussed in this work show the importance of precipitation suppression in MSC in particular and the potential benefit of utilizing explainable machine learning to analyze ACI. By isolating the impact of individual parameters from other confounding factors—which proved to be a challenging endeavour in previous work—the displayed approach opens an avenue to further disentangle the effects of ACI using models or satellite observations.

**Author Contributions:** Conceptualization, L.Z. and H.A.; methodology, L.Z. and H.A.; software, L.Z.; formal analysis, L.Z.; writing—original draft preparation, L.Z.; writing—review and editing, H.A., D.P.G. and J.C.; visualization, L.Z.; funding acquisition, H.A. and J.C. All authors have read and agreed to the published version of the manuscript.

**Funding:** This work has received funding from the European Union's Horizon 2020 research and innovation program under grant agreement no. 821205 (FORCeS). D.P.G. acknowledges support from the Centre for Environmental Modelling And Computation (CEMAC) at the University of Leeds.

**Institutional Review Board Statement:** Not applicable.

**Informed Consent Statement:** Not applicable.

**Data Availability Statement:** Publicly available datasets were analyzed in this study. The C3M dataset can be found at the NASA Langley Research Center: https://doi.org/10.5067/AQUA/CERES/NEWS_CCCM-FM3-MODIS-CAL-CS_L2.RELB1 (accessed on: 12 February 2020). ERA5 data on single levels and on pressure levels are available at the Copernicus Climate Change Service (C3S) Climate Date Store: Single levels—https://doi.org/10.24381/cds.adbb2d47 (accessed on: 8 March 2021), Pressure levels—https://doi.org/10.24381/cds.bd0915c6 (accessed on: 8 March 2021). The AMSR-E dataset is available at the NASA National Snow and Ice Data Center: https://doi.org/10.5067/AMSR-E/AE_RAIN.003 (accessed on: 25 September 2021).

**Acknowledgments:** The authors express their gratitude to two anonymous reviewers for their constructive comments, which have helped improve the manuscript.

**Conflicts of Interest:** The authors declare no conflicts of interest.

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
