# Peer review of "How Cloud Droplet Number Concentration Impacts Liquid Water Path and Precipitation in Marine Stratocumulus Clouds—A Satellite-Based Analysis Using Explainable Machine Learning"

_atmosphere, doi:10.3390/atmos15050596_

Round 1

Reviewer 1 Report

Comments and Suggestions for Authors

Review of the manuscript “atmosphere-2920198” entitled “How cloud droplet number concentration impacts liquid water path and precipitation in marine stratocumulus clouds - a satellite-based analysis using explainable machine learning” written by Zipfel, Anderson, Grosvenor, and Cermak.

The authors of this manuscript obtained cloud-related properties such as LWP and Nd on the five well-known marine stratocumulus regions using satellite observations as well as ERA5, and trained the GBRT model. And then, they utilized SHAP method to explain the contribution of Nd on LWP and precipitation fraction (PF).

The idea of this study is quite interesting. However, the authors make some severe flaws in interpreting the results. The most important thing is that the decrease in Nd-PF SHAP values with increasing Nd does NOT necessarily mean that “Nd reduces PF”. Since the SHAP value simply shows the contribution, the result should be interpreted that “the contribution of Nd to PF is decreasing with increasing Nd”. The statement “Nd reduces PF” can only be validated if the SHAP values of Nd-PF become negative. Moreover, the authors insist clear separation of the SHAP values of Nd-PF values depending on LWP, but the number of such cases are relatively rare; in other words, the SHAP values exhibit almost the same trend in most observations.

Based on this, I recommend rejection on this manuscript.

Reviewer 2 Report

Comments and Suggestions for Authors

Please find the attached Review file.

Round 2

Reviewer 2 Report

Comments and Suggestions for Authors

The manuscript has been significantly revised and improved. The authors have generally addressed all comments successfully. Therefore, I suggest accepting the paper.